# Differential Equation Scaling Limits of Shaped and Unshaped Neural Networks

**Mufan (Bill) Li**                                                    *mufan.li@princeton.edu*
*Princeton University*

**Mihai Nica**                                                         *nicam@uoguelph.ca*
*University of Guelph and Vector Institute*

**Reviewed on OpenReview:** *https://openreview.net/forum?id=iRDwUXYsSJ*

## Abstract

Recent analyses of neural networks with *shaped* activations (i.e. the activation function is scaled as the network size grows) have led to scaling limits described by differential equations. However, these results do not a priori tell us anything about "ordinary" unshaped networks, where the activation is unchanged as the network size grows. In this article, we find similar differential equation based asymptotic characterization for two types of unshaped networks.

- Firstly, we show that the following two architectures converge to the same infinite-depth-and-width limit at initialization: (i) a fully connected ResNet with a $d^{-1/2}$ factor on the residual branch, where $d$ is the network depth. (ii) a multilayer perceptron (MLP) with depth $d \ll$ width $n$ and shaped ReLU activation at rate $d^{-1/2}$.
- Secondly, for an unshaped MLP at initialization, we derive the first order asymptotic correction to the layerwise correlation. In particular, if $\rho_\ell$ is the correlation at layer $\ell$, then $q_t = \ell^2(1 - \rho_\ell)$ with $t = \frac{\ell}{n}$ converges to an SDE with a singularity at $t = 0$.

These results together provide a connection between shaped and unshaped network architectures, and opens up the possibility of studying the effect of normalization methods and how it connects with shaping activation functions.

## 1 Introduction

Martens et al. (2021); Zhang et al. (2022) proposed transforming the activation function to be more linear as the neural network becomes larger in size, which significantly improved the speed of training deep networks without batch normalization. Based on the infinite-depth-and-width limit analysis of Li et al. (2022), the principle of these transformations can be roughly summarized as follows: choose an activation function $\varphi_s : \mathbb{R} \to \mathbb{R}$ as a perturbation of the identity map depending on the network width $n$ (or depth $d = n^{2p}, p > 0$)

$$\varphi_s(x) = x + \frac{1}{n^p}h(x) + O(n^{-2p}) = x + \frac{1}{\sqrt{d}}h(x) + O(d^{-1}), \tag{1.1}$$

where for simplicity we will ignore the higher order terms for now. Li et al. (2022) also showed the limiting multilayer perceptron (MLP) can be described by a Neural Covariance stochastic differential equation (SDE). Furthermore, it appears the choice of $p = \frac{1}{2}$ is necessary to reach a non-degenerate nor trivial limit, at least when the depth-to-width ratio $\frac{d}{n}$ converges to a positive constant (see (Li et al., 2022, Proposition 3.4)).

Recently, Hayou & Yang (2023); Cirone et al. (2023) also studied the infinite-depth-and-width limit of a specific ResNet architecture (He et al., 2016). Most interestingly, the limit is described by an ordinary differential equation (ODE) very similar to the neural covariance SDE. Furthermore, Hayou & Yang (2023) showed the width and depth limits commute, i.e. no dependence on the depth-to-width ratio $\frac{d}{n}$. It is then natural to consider a more careful comparison and understanding of the two differential equations.

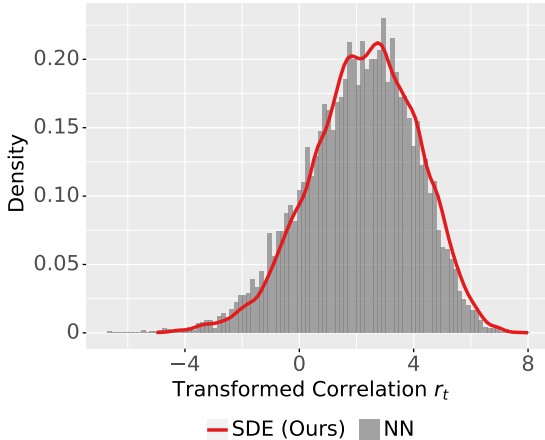

Figure 1: Empirical distribution of the transformed correlation $r_t = \log(\ell^2(1 - \rho_\ell))$ for an unshaped ReLU MLP, SDE sample density computed via kernel density estimation. Simulated with $n = d = 150, \rho_0 = 0.3, r_0 = \log(1 - \rho_0) = \log(0.7)$, SDE step size $10^{-2}$, and $2^{13}$ samples.

At the same time, Li et al. (2022) demonstrated the unshaped network also has an accurate approximation via a Markov chain. Jakub & Nica (2023) further studied the large width asymptotics of the Markov chain updates, where the transition kernel still depends on the width. Since the Markov chain quickly converges to a fixed point, it does not immediately appear to have a scaling limit. However, this motivates us to consider a modified scaling around the fixed point, so that we can recover a first order asymptotic correction term.

In this note, we provide two technical results that address both of the above questions. Both of the results are achieved by considering a modification of the scaling, which leads to the following results.

- Firstly, we demonstrate that shaping the activation has a close connection to ResNets, and the covariance ODE is in fact just the deterministic drift component of the covariance SDE. Furthermore, in the limit as the scaled ratio of $\frac{d}{n^{2p}}$ to converges to a positive constant and $p \in (0, \frac{1}{2})$, the shaped MLP covariance also converges to the same ODE.

- Secondly, we analyze the correlation of an *unshaped* MLP, providing a derivation of the first order asymptotic correction. The correction term arises from rescaling the correlation $\rho_\ell$ in layer $\ell$ by $q_\ell = \ell^2(1 - \rho_\ell)$, and we show it is closely approximated by an SDE.

The rest of this article is organized as follows. Firstly, we will provide a brief literature review in the rest of this section. Next, we will review the most relevant known results on this covariance SDEs and ODEs in Section 2. Then in Section 3, we will make the connection between shaping and ResNets precise. At the same time, we will also provide a derivation of the unshaped regime in Section 4, where we show that by modifying the scaling yet again, we can recover another SDE related to the correlation of a ReLU MLP.

## 1.1 Related Work

On a conceptual level, the main difficulty of analyzing neural networks is due to the lack of mathematical tractability. In a seminal work, Neal (1995) showed that two layer neural networks at initialization converges to a Gaussian process. Beyond the result itself, the conceptual breakthrough opened up the field to analyzing large size asymptotics of neural networks. In particular, this led to a large body of work on large or infinite width neural networks (Lee et al., 2018; Jacot et al., 2018; Du et al., 2019; Mei et al., 2018; Sirignano & Spiliopoulos, 2018; Yang, 2019; Bartlett et al., 2021). However, majority of these results relied on the network converging to a kernel limit, which are known to perform worse than neural networks (Ghorbani et al., 2020). The gap in performance is believed to be primarily due to a lack of feature learning (Yang & Hu, 2021; Abbe

| Notation | Description | Notation | Description | Table 1: Notation |
|---|---|---|---|---|
| $n_{\text{in}} \in \mathbb{N}$ | Input dimension | $n_{\text{out}} \in \mathbb{N}$ | Output dimension | |
| $n \in \mathbb{N}$ | Hidden layer width | $d \in \mathbb{N}$ | Number of hidden layers (depth) | |
| $\varphi(\cdot)$ | Base activation | $\varphi_s(\cdot)$ | Shaped activation | |
| $x^\alpha \in \mathbb{R}^{n_{\text{in}}}$ | Input for $1 \leq \alpha \leq m$ | $W_0 \in \mathbb{R}^{n_{\text{in}} \times n}$ | Weight matrix at layer 0 | |
| $z_{\text{out}}^\alpha \in \mathbb{R}^{n_{\text{out}}}$ | Network output | $W_{\text{out}} \in \mathbb{R}^{n \times n_{\text{out}}}$ | Weight matrix at final layer | |
| $z_\ell^\alpha \in \mathbb{R}^n$ | Neurons (pre-activation) | $W_\ell \in \mathbb{R}^{n \times n}$ | Weight matrix at layer $1 \leq \ell \leq d$ | |
| | for layer $1 \leq \ell \leq d$ | | **All weights initialized iid** $\sim \mathcal{N}(0,1)$ | |
| $\varphi_\ell^\alpha \in \mathbb{R}^n$ | Neurons (post-activation) | $c \in \mathbb{R}$ | Normalizing constant | |
| | for layer $1 \leq \ell \leq d$ | | $c := \left( \mathbb{E}\, \varphi(g)^2 \right)^{-1}$ for $g \sim \mathcal{N}(0,1)$ | |
| $V_\ell^{\alpha\beta} \in \mathbb{R}$ | Covariance $\frac{c}{n}\langle \varphi_\ell^\alpha, \varphi_\ell^\beta \rangle$ | $\rho_\ell^{\alpha\beta} \in [-1,1]$ | Correlation $V_\ell^{\alpha\beta}/\sqrt{V_\ell^{\alpha\alpha} V_\ell^{\beta\beta}}$ | |

et al., 2022; Ba et al., 2022). While this motivated the study of several alternative scaling limits, in this work we are mostly interested in the infinite-depth-and-width limit.

First investigated by Hanin & Nica (2019b), it was shown that not only does this regime not converge to a Gaussian process at initialization, it also learns features Hanin & Nica (2019a). This limit has since been analyzed with transform based methods (Noci et al., 2021) and central limit theorem approaches (Li et al., 2021). As we will describe in more detail soon, the result of most interest is the covariance SDE limit of Li et al. (2022). The MLP results were also further extended to the transformer setting (Noci et al., 2023).

The $d^{-1/2}$ scaling for ResNets was first considered by Hayou et al. (2021), with the depth limit carefully studied afterwards (Hayou, 2022; Hayou & Yang, 2023; Hayou, 2023). Fischer et al. (2023) also arrived at the same scaling through a different theoretical approach. This scaling has found applications for hyperparameter tuning (Bordelon et al., 2023; Yang et al., 2023) when used in conjunction with the $\mu$P scaling (Yang & Hu, 2021).

Batch and layer normalization methods were introduced as a remedy for unstable training (Ioffe & Szegedy, 2015; Ba et al., 2016), albeit theoretical analyses of these highly discrete changes per layer has been challenging. A recent promising approach studies the isometry gap, and shows that batch normalization methods achieves a similar effect as shaping activation functions (Meterez et al., 2023). Theoretical connections between these approaches using a differential equation based description remains an open problem.

## 2 Background on Shaped Networks and ResNets

Let $\{x^\alpha\}_{\alpha=1}^m$ be a set of input data points in $\mathbb{R}^{n_{\text{in}}}$, and let $z_\ell^\alpha \in \mathbb{R}^n$ denote the $\ell$-th hidden layer with respect to the input $x^\alpha$. We consider the standard width-$n$ depth-$d$ MLP architecture with He-initialization (He et al., 2015) defined by the following recursion

$$z_{\text{out}}^\alpha = \sqrt{\frac{c}{n}} W_{\text{out}}\, \varphi(z_d^\alpha), \quad z_{\ell+1}^\alpha = \sqrt{\frac{c}{n}} W_\ell\, \varphi_s(z_\ell^\alpha), \quad z_1^\alpha = \frac{1}{\sqrt{n_{\text{in}}}} W_{\text{in}}\, x^\alpha, \tag{2.1}$$

where $\varphi_s : \mathbb{R} \to \mathbb{R}$ is the activation function to be specified, $c^{-1} = \mathbb{E}\, \varphi_s(g)^2$ for $g \sim N(0,1)$, $z_\ell^\alpha \in \mathbb{R}^n, z_{\text{out}}^\alpha \in \mathbb{R}^{n_{\text{out}}}$, and the matrices $W_{\text{out}} \in \mathbb{R}^{n_{\text{out}} \times n}, W_\ell \in \mathbb{R}^{n \times n}, W_{\text{in}} \in \mathbb{R}^{n \times n_{\text{in}}}$ are initialized with iid $N(0,1)$ entries.

The main structure used to study neural networks at initialization, such as the neural network Gaussian processes (NNGP) (Neal, 1995; Lee et al., 2018), is on the conditional Gaussian property. More precisely, if we condition on the previous layers $\mathcal{F}_\ell = \sigma((z_k^\alpha)_{\alpha \in [m], k \leq \ell})$, we have that

$$[z_{\ell+1}^\alpha]_{\alpha=1}^m | \mathcal{F}_\ell \overset{d}{=} \mathcal{N}\left(0, \frac{c}{n}[\langle \varphi_\ell^\alpha, \varphi_\ell^\beta \rangle]_{\alpha,\beta=1}^m \otimes I_n \right), \tag{2.2}$$

where we use the notation $[\cdot]_{\alpha=1}^m$ to vertically stack vectors, let $\varphi_\ell^\alpha = \varphi_s(z_\ell^\alpha)$ be the post activation hidden layer, and let $\otimes$ be the Kronecker product.

This naturally leads us to define the covariance matrix as $V_\ell := \frac{c}{n}[\langle \varphi_\ell^\alpha, \varphi_\ell^\beta \rangle]_{\alpha,\beta=1}^m$. The NNGP results essentially reduces down to applying the Law of Large Numbers inductively to show the covariance $V_\ell^{\alpha\beta}$ converges to its expect value. More precisely, if $[z_\ell^\alpha]_{\alpha=1}^m \sim \mathcal{N}(0, V_{\ell-1} \otimes I_n)$, then in the limit as $n \to \infty$

$$V_\ell^{\alpha\beta} = \frac{c}{n} \sum_{i=1}^n \varphi_s(z_{\ell,i}^\alpha) \varphi_s(z_{\ell,i}^\beta) \xrightarrow{d} c\, \mathbb{E}\, \varphi_s(g^\alpha) \varphi_s(g^\beta)\,, \quad \text{where } [g^\alpha, g^\beta]^\top \sim \mathcal{N}\left(0, \begin{bmatrix} V_{\ell-1}^{\alpha\alpha} & V_{\ell-1}^{\alpha\beta} \\ V_{\ell-1}^{\alpha\beta} & V_{\ell-1}^{\beta\beta} \end{bmatrix}\right)\,. \quad (2.3)$$

However, since the next layer covariance $V_\ell$ is a deterministic function of the previous $V_{\ell-1}$, this forms a fixed point type iteration $V_\ell = f(V_{\ell-1})$. Indeed, if we observe the correlations $\rho_\ell^{\alpha\beta} = \frac{\langle \varphi_\ell^\alpha, \varphi_\ell^\beta \rangle}{|\varphi_\ell^\alpha||\varphi_\ell^\beta|}$, which is bounded in $[-1, 1]$, it does in fact converge to a fixed point at $\rho_\infty = 1$ for ReLU activations (see e.g. Proposition 3.4 (i) Li et al. (2022)).

This degeneracy of correlations also causes unstable gradients, which led to a proposal by Martens et al. (2021); Zhang et al. (2022) to modify the shape of activation functions $\varphi_s$ depending on the size of the network, leading to improved training speeds without using normalization methods. However, as both of these works computed the activation shape based on a set of criteria, it is unclear what is the appropriate modification in the scaling limit as depth and width both approaches infinity.

## 2.1 Shaped Limit of Neural Networks

To this end, Li et al. (2022) is the first to describe the limit as $d, n \to \infty$ with $\frac{d}{n} \to T > 0$ and the activation function $\varphi_s$ is shaped at a precise rate to be closer to the identity as we increase $n$. In particular, the covariance matrix $V_\ell$ forms a Markov chain, as it satisfies the definition of $V_{\ell+1}|\mathcal{F}_\ell = V_{\ell+1}|\sigma(V_\ell)$ (see e.g. (2.3)). The main result describes the scaling limit of the Markov chain $V_{\lfloor tn \rfloor}$ via a stochastic differential equation (SDE), which we can intuitively interpret as the Euler discretization converging to the differential equation

$$V_{\ell+1} = V_\ell + \frac{b(V_\ell)}{n} + \frac{\Sigma(V_\ell)^{1/2} \xi_\ell}{\sqrt{n}} + O(n^{-3/2}) \xrightarrow{n \to \infty} dV_t = b(V_t)\, dt + \Sigma(V_t)\, dB_t\,, \quad (2.4)$$

where $\xi_\ell$ are iid. zero mean identity variance random vectors, and we interpret $\frac{1}{n}$ as the step size of the discretization.

The activation functions are modified as follows. For a ReLU-like activation, we choose

$$\varphi_s(x) = s_+ \max(x, 0) + s_- \min(x, 0)\,, \quad s_\pm = 1 + \frac{c_\pm}{n^p}\,, c_\pm \in \mathbb{R}, p \geq 0\,, \quad (2.5)$$

or for a smooth activation $\varphi \in C^4(\mathbb{R})$ such that $\varphi(0) = 0, \varphi'(0) = 1$ and $\varphi^{(4)}(x)$ bounded by a polynomial, we choose

$$\varphi_s(x) = s\, \varphi\left(\frac{x}{s}\right)\,, \quad s = an^p, a \neq 0, p \geq 0\,. \quad (2.6)$$

We will first recall one of the main results of Li et al. (2022), which is stated informally[1] below.

**Theorem 2.1** (Theorem 3.2 and 3.9 of Li et al. (2022), Informal). *Let $p = \frac{1}{2}$. Then in the limit as $d, n \to \infty, \frac{d}{n} \to T > 0$, and $\varphi_s$ defined as above, we have that the upper triangular entries of $V_{\lfloor tn \rfloor}$ (flattened to a vector) converges to the following SDE weakly*

$$dV_t = b(V_t)\, dt + \Sigma(V_t)^{1/2}\, dB_t\,, \quad V_0 = \frac{1}{n_{in}}[\langle x^\alpha, x^\beta \rangle]_{\alpha,\beta=1}^m\,, \quad (2.7)$$

*where $\Sigma(V)|_{\alpha\beta,\gamma\delta} = V^{\alpha\gamma}V^{\beta\delta} + V^{\alpha\delta}V^{\beta\gamma}$, and if $\varphi$ is a ReLU-like activation we have*

$$b(V)|_{\alpha\beta} = \nu(\rho^{\alpha\beta})\sqrt{V^{\alpha\alpha}V^{\beta\beta}}\,, \quad \rho^{\alpha\beta} = \frac{V^{\alpha\beta}}{\sqrt{V^{\alpha\alpha}V^{\beta\beta}}}\,, \quad \nu(\rho) = \frac{(c_+ - c_-)^2}{2\pi}\left(\sqrt{1 - \rho^2} - \rho \arccos \rho\right)\,, \quad (2.8)$$

---

[1]The statement is "informal" in the sense that we have stated what the final limit is, but not the precise sense of the convergence. See Appendix A for a rigorous treatment of the convergence result.

*or else if $\varphi$ is a smooth activation we have*

$$b^{\alpha\beta}(V_t) = \frac{\varphi''(0)^2}{4a^2}\left(V_t^{\alpha\alpha}V_t^{\beta\beta} + V_t^{\alpha\beta}(2V_t^{\alpha\beta} - 3)\right) + \frac{\varphi'''(0)}{2a^2}V_t^{\alpha\beta}(V_t^{\alpha\alpha} + V_t^{\beta\beta} - 2). \qquad (2.9)$$

While the formulae may seem overwhelming, there is actually a fairly straight forward interpretation of both the drift $b$ and the diffusion coefficient $\Sigma^{1/2}$. In particular, for the unshaped ReLU network, that is $\varphi_s(x) = \max(x, 0)$, the deterministic component of the update for the covariance matrix compared to the shaped network are as follows

$$\text{Unshaped: } \mathbb{E}\,V_{\ell+1} - V_\ell \propto b(V_\ell),$$

$$\text{Shaped: } \mathbb{E}\,V_{\ell+1} - V_\ell \propto \frac{b(V_\ell)}{n}. \qquad (2.10)$$

Effectively, the drift component got slowed down by a multiplicative factor of $\frac{1}{n}$, which can interpreted as an Euler discretization step. In order to achieve a stable limit as we take depth $d$ to infinity, we would also require each layer to contribute infinitesimally as a rate proportional to $\frac{1}{d}$, therefore this is a desired rescaling.

The diffusion coefficient is much more interesting. Since we can interpret this diffusion to be on the manifold of symmetric positive definite matrices, we would expect the diffusion coefficient of Brownian motions on this manifold to correspond to a Riemannian metric. Indeed, this is shown in Li et al. (2024), as $\Sigma^{-1}$ corresponds to the affine invariant metric. More precisely, for all $V \in \text{SPD}(m)$, the inner product corresponding to $\Sigma^{-1}$

$$\langle A, B \rangle_{\Sigma^{-1}(V)} = \sum_{1 \le i \le j \le m} A_{ij} B_{ij} \Sigma^{-1}(V)_{ij} = \frac{1}{2}\text{Tr}(AV^{-1}BV^{-1}), \quad \text{for all } A, B \in \text{Sym}(m). \qquad (2.11)$$

Furthermore, this gives the linear network SDE $dV_t = \Sigma(V_t)^{1/2}\,dB_t$ an interpretation as the dual Brownian motion in information geometry, which uses a pair of dually flat affine connections instead of the standard Levi-Civita connection.

## 2.2 ODE Limit of Residual Networks

At the same time, Hayou & Yang (2023); Cirone et al. (2023) found an ordinary differential equation (ODE) limit describing the covariance matrix for infinite-depth-and-width ResNets. The authors considered a ResNet architecture with a $\frac{1}{\sqrt{d}}$ factor on their residual branch, more precisely their recursion is defined as follows (in our notation and convention)

$$z_{\ell+1}^\alpha = z_\ell^\alpha + \frac{1}{\sqrt{dn}}W_\ell\,\varphi(z_\ell^\alpha), \quad \text{where } \varphi(x) = \max(x, 0). \qquad (2.12)$$

Intuitively, this is similar to shaping activations, it also weakens the effect of each layer as we take $d$ to infinity. This will be discussed in more details in the following section.

One of their main results can be stated informally as follows.

**Theorem 2.2** (Theorem 2 of Hayou & Yang (2023), Informal). *Let $d, n \to \infty$ (in any order), and the covariance process $V_{\lfloor td \rfloor}^{\alpha\beta}$ converges to the following ODE*

$$\frac{d}{d_t}V_t^{\alpha\beta} = \frac{1}{2}\frac{f(\rho_t^{\alpha\beta})}{\rho_t^{\alpha\beta}}V_t^{\alpha\beta}, \qquad (2.13)$$

*where $f(\rho) = \frac{1}{\pi}(\rho\arcsin\rho + \sqrt{1 - \rho^2}) + \frac{1}{2}\rho$.*

Here, we observe this ODE (2.13) is exactly the drift component of the covariance SDE (2.7), i.e.

$$\frac{1}{2}\frac{f(\rho_t^{\alpha\beta})}{\rho_t^{\alpha\beta}}V_t^{\alpha\beta} = \nu(\rho_t^{\alpha\beta})\sqrt{V_t^{\alpha\alpha}V_t^{\beta\beta}}, \quad \text{if } (c_+ - c_-)^2 = 1, \frac{d}{n} = 1. \qquad (2.14)$$

To see this, we just need to use the identity $\arcsin \rho = \frac{\pi}{2} - \arccos \rho$ to get that $\frac{1}{2} f(\rho) = \nu(\rho)$, and that $\frac{V_t^{\alpha\beta}}{\rho_t^{\alpha\beta}} = \sqrt{V_t^{\alpha\alpha} V_t^{\beta\beta}}$ is exactly the definition of $\rho_t^{\alpha\beta}$. We also note the correlation ODE $\frac{d}{dt} \rho_t^{\alpha\beta} = \nu(\rho_t^{\alpha\beta})$ was first derived in (Zhang et al., 2022, Proposition 3), where they considered the sequential width then depth limit with a fixed initial and terminal condition.

In the next section, we will describe another way to recover this ODE from an alternative scaling limit of the shaped MLP.

## 3  An Alternative Shaped Limit for $p \in (0, \frac{1}{2})$

We start by providing some intuitions on this result. The shaped MLP can be seen as a layerwise perturbation of the linear network

$$z_{\ell+1} = \sqrt{\frac{c}{n}} W_\ell \, \varphi_s(z_\ell) \approx \sqrt{\frac{c}{n}} W_\ell \, z_\ell + \frac{1}{\sqrt{d}} \sqrt{\frac{c}{n}} W_\ell \, h(z_\ell) \,, \tag{3.1}$$

where $c^{-1} = \mathbb{E}\, \varphi_s(g)^2$ for $g \sim N(0,1)$ corresponds to the He-initialization (He et al., 2015), and $W_\ell \in \mathbb{R}^{n \times n}$ has iid $N(0,1)$ entries.

On an intuitive level (which we will make precise in Remark 3.3), if we take the infinite-width limit first, then this removes the effect of the random weights. In other words, if we replace the weights $\frac{1}{\sqrt{n}} W_\ell$ with the identity matrix $I_n$ in each hidden layer, we get the same limit at initialization. Therefore, we can heuristically write

$$z_{\ell+1} \approx z_\ell + \frac{1}{\sqrt{d}} h(z_\ell) \,, \tag{3.2}$$

where we also used the fact $c \to 1$ in the limit.

Observe this is resembling a ResNet, where the first $z_\ell$ is the skip connection. In fact, we can again heuristically add back in the weights on the residual branch to get

$$z_{\ell+1} \approx z_\ell + \frac{1}{\sqrt{d}} W_\ell h(z_\ell) \,, \tag{3.3}$$

which exactly recovers the ResNet formulation of Hayou et al. (2021); Hayou & Yang (2023), where the authors studied the case when $h(x) = \max(x, 0)$ is the ReLU activation.

*Remark* 3.1. On a heuristic level, this implies that whenever the width limit is taken first (or equivalently $d = n^{2p}$ for $p \in (0, 1/2)$), the shaped network with shaping parameter $d^{-1/2}$ has *the same limiting distribution at initialization* as a ResNet with a $d^{-1/2}$ weighting on the residual branch.

However, we note that despite having identical ODE for the covariance at initialization, *this does not imply the training dynamics will be the same* — it will likely be different. Furthermore, since Hayou & Yang (2023) showed the width and depth limits commute for ResNets, this provides the additional insight that noncommunitativity of limits in shaped MLPs arises from the product of random matrices.

### 3.1  Precise Results

The core object that forms a Markov chain is the post-activation covariance matrix $\frac{c}{n} [\langle \varphi_\ell^\alpha, \varphi_\ell^\beta \rangle]_{\alpha,\beta=1}^m$. To see this, we will use the property of Gaussian matrix multiplication, where we let $W \in \mathbb{R}^{n \times n}$ with iid entries $W_{ij} \sim N(0,1)$, and $\{u^\alpha\}_{\alpha=1}^m \in \mathbb{R}^n$ be a collection of constant vectors, which gives us

$$[W u^\alpha]_{\alpha=1}^m \overset{d}{=} N(0, [\langle u^\alpha, u^\beta \rangle]_{\alpha=1}^m \otimes I_n) \,, \tag{3.4}$$

where we use the notation $[v^\alpha]_{\alpha=1}^m$ to stack the vectors vertically. This forms a Markov chain because we can condition on $\mathcal{F}_\ell = \sigma([z_\ell^\alpha]_{\alpha=1}^m)$ to get

$$[z_{\ell+1}^\alpha]_{\alpha=1}^m | \mathcal{F}_\ell = [z_{\ell+1}^\alpha]_{\alpha=1}^m | \sigma(V_\ell) \sim N(0, V_\ell \otimes I_n) \,. \tag{3.5}$$

and we can see that $V_{\ell+1} | \mathcal{F}_\ell = V_{\ell+1} | \sigma(V_\ell)$, which is exactly the definition of a Markov chain.

We will start by deriving the precise Markov chain update up to a term of size $O(n^{-3p})$, which will be a slight modification of the Euler discretization we saw in (2.4).

**Lemma 3.2** (Covariance Markov Chain for the Shaped MLP). *Let $z_\ell^\alpha$ be the MLP in defined in (2.1) with shaped ReLU activations defined in (2.5). For $p \in (0, \frac{1}{2})$ and $d = n^{2p}$, the Markov chain satisfies*

$$V_{\ell+1} = V_\ell + \frac{b(V_\ell)}{d} + \frac{\Sigma_s(V_\ell)^{1/2}\,\xi_\ell}{\sqrt{n}} + O(d^{-3/2})\,, \tag{3.6}$$

*where $\{\xi_\ell\}_{\ell \geq 0}$ are iid zero mean and identity covariance random vectors, and in the limit as $n, d \to \infty$ we have that $\Sigma_s \to \Sigma$, and the coefficients are defined as*

$$b(V)^{\alpha\beta} = \nu(\rho^{\alpha\beta})\sqrt{V^{\alpha\alpha}V^{\beta\beta}}\,, \quad \Sigma(V)^{\alpha\beta,\gamma\delta} = V^{\alpha\gamma}V^{\beta\delta} + V^{\alpha\delta}V^{\beta\gamma}\,, \tag{3.7}$$

*where $\rho^{\alpha\beta} = \frac{V^{\alpha\beta}}{\sqrt{V^{\alpha\alpha}V^{\beta\beta}}}$, and $\nu(\rho) = \frac{(c_+ - c_-)^2}{2\pi}\left(\sqrt{1-\rho^2} + \rho\arccos\rho\right)$.*

*Proof.* To start, we will observe that conditioned on $\mathcal{F}_\ell$, we have that

$$V_{\ell+1}^{\alpha\beta}|\mathcal{F}_\ell = \frac{c}{n}\sum_{i=1}^n \varphi_s(z_{\ell+1,i}^\alpha)\,\varphi_s(z_{\ell+1,i}^\beta) = |\varphi_\ell^\alpha|\,|\varphi_\ell^\beta|\frac{c}{n}\sum_{i=1}^n \varphi_s(g_i^\alpha)\,\varphi_s(g_i^\beta)\,, \tag{3.8}$$

where used the fact that $z_{\ell+1}$ are jointly Gaussian, and we have that

$$\begin{bmatrix} g_i^\alpha \\ g_i^\beta \end{bmatrix}_{i=1}^n \sim \mathcal{N}\left(0, \begin{bmatrix} 1 & \rho_\ell^{\alpha\beta} \\ \rho_\ell^{\alpha\beta} & 1 \end{bmatrix} \otimes I_n\right)\,, \tag{3.9}$$

for $\rho^{\alpha\beta} = \frac{V^{\alpha\beta}}{\sqrt{V^{\alpha\alpha}V^{\beta\beta}}}$. At this point, we can define $K_1(\rho^{\alpha\beta}) = \mathbb{E}\,\varphi_s(g_i^\alpha)\,\varphi_s(g_i^\beta)$ and the random variable

$$R_\ell^{\alpha\beta} = \frac{1}{\sqrt{n}}\sum_{i=1}^n c\varphi_s(g_i^\alpha)\varphi_s(g_i^\beta) - cK_1(\rho_\ell^{\alpha\beta})\,, \tag{3.10}$$

which allows us to decompose this into a deterministic and a random component as

$$V_{\ell+1}^{\alpha\beta}|\mathcal{F}_\ell = |\varphi_\ell^\alpha|\,|\varphi_\ell^\beta|\left(cK_1(\rho_\ell^{\alpha\beta}) + \frac{1}{\sqrt{n}}R_\ell^{\alpha\beta}\right)\,. \tag{3.11}$$

We can Taylor expand $cK_1(\rho)$ in terms of $n^{-p}$ to get the following result from Lemma B.1

$$cK_1(\rho) = \rho + \frac{\nu(\rho)}{n^{2p}} + O(n^{-3p})\,, \quad \nu(\rho) = \frac{(c_+ - c_-)^2}{2\pi}\left(\sqrt{1-\rho^2} + \rho\arccos\rho\right)\,. \tag{3.12}$$

Similarly from Lemma B.3, we also have the approximation

$$\mathbb{E}\,R_\ell^{\alpha\beta}R_\ell^{\gamma\delta} = \rho_\ell^{\alpha\gamma}\rho_\ell^{\beta\delta} + \rho_\ell^{\alpha\delta}\rho_\ell^{\beta\gamma} + O(n^{-p})\,. \tag{3.13}$$

Putting everything together, we can write

$$cK_1(\rho_\ell^{\alpha\beta}) + \frac{1}{\sqrt{n}}R_\ell^{\alpha\beta} = \rho_\ell^{\alpha\beta} + \frac{\nu(\rho_\ell^{\alpha\beta})}{n^{2p}} + \frac{1}{\sqrt{n}}R_\ell^{\alpha\beta} + O(n^{-3p})\,, \tag{3.14}$$

which implies we can write

$$V_{\ell+1}^{\alpha\beta} = V_\ell^{\alpha\beta} + \frac{b(V_\ell)^{\alpha\beta}}{n^{2p}} + \frac{R^{\alpha\beta_\ell}}{\sqrt{n}} + O(n^{-3p})\,, \tag{3.15}$$

where $b(V) = \nu(\rho_\ell^{\alpha\beta})\sqrt{V_\ell^{\alpha\alpha}V_\ell^{\beta\beta}}$. Now taking the upper triangular entries of $V_\ell$ as a vector, we have that

$$V_{\ell+1} = V_\ell + \frac{b(V_\ell)}{n^{2p}} + \frac{\Sigma_s(V_\ell)\xi_\ell}{\sqrt{n}} + O(n^{-3p})\,, \tag{3.16}$$

where $\Sigma_s(V)|_{\alpha\beta,\gamma\delta} = \sqrt{V^{\alpha\alpha}V^{\beta\beta}}(\rho^{\alpha\gamma}\rho^{\beta\delta} + \rho^{\alpha\delta}\rho^{\beta\gamma}) + O(n^{-p}) = V^{\alpha\gamma}V^{\beta\delta} + V^{\alpha\delta}V^{\beta\gamma} + O(n^{-p})$, and $\xi_\ell$ is a zero mean identity covariance random vector. We recover the exact desire result from writing $d = n^{2p}$.

$\square$

*Remark* 3.3. We note that the drift term arising from the activation depends only on the depth $d$ and the random term only depends on the width $n$. If we decouple the dependence on $d$ and $n$, and take the infinite-width limit first, we arrive at

$$V_{\ell+1} = V_\ell + \frac{b_s(V_\ell)}{d} + O(n^{-3/2}), \tag{3.17}$$

which is equivalent to removing the randomness of the weights.

We note this Markov chain behaves like a sum of two Euler updates with step sizes $\frac{1}{n^{2p}}$ and $\frac{1}{n}$, where the $\frac{1}{n}$ corresponds to the random term with coefficient $\frac{1}{\sqrt{n}}$. However since $p \in (0, \frac{1}{2})$, the first term with step size $\frac{1}{n^{2p}}$ will dominate, which is the term that corresponds to shaping the activation function. Therefore, we expect the random term to vanish in the limit, and hence leaving us with an ODE only. We make this result precise in the following Proposition.

**Proposition 3.4** (Covariance ODE for the Shaped ReLU MLP). *Let $p \in (0, \frac{1}{2})$. Then in the limit as $d, n \to \infty, \frac{d}{n^{2p}} \to 1$, and $\varphi_s$ is the shaped ReLU defined in (2.5), we have that the upper triangular entries of $V_{\lfloor tn \rfloor}$ (flattened to a vector) converges to the following ODE weakly with respect to the Skorohod topology of $D_{\mathbb{R}_+, \mathbb{R}^{m(m+1)/2}}$*

$$dV_t = b(V_t)\, dt, \quad V_0 = \frac{1}{n_{in}} [\langle x^\alpha, x^\beta \rangle]_{\alpha,\beta=1}^m, \tag{3.18}$$

*where $b$ is defined in Theorem 2.1 and Lemma 3.2.*

*Proof.* Starting with the Markov chain in Lemma 3.2, we will treat the random term of order $O(n^{-1/2})$ as part of the drift instead. More precisely, we let

$$\frac{\widehat{b}(V)}{n^{2p}} = \frac{b(V)}{n^{2p}} + \frac{\Sigma_s(V)^{1/2}\xi_\ell}{\sqrt{n}}, \tag{3.19}$$

which in expectation just equals to $b(V)n^{-2p}$. Since there is no random term at the order of $n^{-p}$, we can apply Proposition A.7 to the special as if the diffusion coefficient is equal to zero. At the same time, since the higher order term in the Markov chain is at the desired order of $O(n^{-3p})$, which will vanish in the limit, we get the desired result.

$$\square$$

At this point it's worth pointing out the regime of $p \in (0, \frac{1}{2})$ was studied in Li et al. (2022), but the scaling limit was taken to be $\frac{d}{n} \to T$ instead of $\frac{d}{n^{2p}}$. This led to a "degenerate" regime where $\rho_t = 1$ for all $t > 0$. The above ODE result implies that the degenerate limit can be characterized in a more refined way if the scaling is chosen carefully.

In the next and final section, we show that actually even when the network is unshaped (i.e. $p = 0$), there exists a scaling such that we can characterize the limiting Markov chain up to the first order asymptotic correction.

## 4 An SDE for the Unshaped ReLU MLP

In this section, we let $\varphi_s(x) = \varphi(x) = \max(x, 0)$, and we are interested in studying the correlation

$$\rho_\ell^{\alpha\beta} = \frac{V_\ell^{\alpha\beta}}{\sqrt{V_\ell^{\alpha\alpha}V_\ell^{\beta\beta}}} = \frac{\langle \varphi_\ell^\alpha, \varphi_\ell^\beta \rangle}{|\varphi_\ell^\alpha|\,|\varphi_\ell^\beta|}. \tag{4.1}$$

From this point onwards, we will only consider the marginal over two inputs, so we will drop the superscript $\alpha\beta$. Similar to the previous section, we will also start by providing an intuitive sketch.

Many existing work has derived the rough asymptotic order of the unshaped correlation to be $\rho_\ell = 1 - O(\ell^{-2})$, where $\ell$ is the layer (see for example Appendix E of Li et al. (2022) and Jakub & Nica (2023)). Firstly, this

implies that a Taylor expansion of all functions of $\rho$ in the Markov chain update around $\rho = 1$ will be very accurate. At the same time, it is natural to magnify the object inside the big $O$ by reverting the scaling, or more precisely consider the object

$$q_\ell = \ell^2 (1 - \rho_\ell), \tag{4.2}$$

which will hopefully remain at size $\Theta(1)$.

For simplicity, we can consider the infinite-width update of the unshaped correlation (which corresponds to the zeroth-order Taylor expansion in $\frac{1}{n}$)

$$\rho_{\ell+1} = \rho_\ell + c_1 (1 - \rho_\ell)^{3/2} + O((1 - \rho_\ell)^{5/2}), \tag{4.3}$$

where for the sake of illustration we will take $c_1 = 1$ and drop the big $O$ term for now. Substituting in $q_\ell$, we can recover the update

$$q_{\ell+1} = q_\ell + \frac{2q_\ell}{\ell} - \frac{q_\ell^{3/2}}{\ell}. \tag{4.4}$$

While this doesn't quite look like an Euler update just yet, we can substitute in $t = \frac{\ell}{n}$ for the time scale, which will lead us to have

$$q_{\ell+1} = q_\ell + \frac{1}{tn} \left( 2q_\ell - q_\ell^{3/2} \right), \tag{4.5}$$

hence (heuristically) giving us the singular ODE

$$dq_t = \frac{2q_t - q_t^{3/2}}{t}. \tag{4.6}$$

To recover the SDE, we will simply include the additional terms of the Markov chain instead taking the infinite-width limit first.

## 4.1 Full Derivation

In the rest of this section, we will provide a derivation of an SDE arising from an appropriate scaling of $\rho_\ell$.

**Theorem 4.1** (Rescaled Correlation). *Let $q_\ell = \ell^2 (1 - \rho_\ell)$. Then for all $t_0 > 0$, the process $\{q_{\lfloor tn \rfloor}\}_{t \geq t_0}$ converges to a solution of the following SDE weakly in the Skorohod topology (see (Li et al., 2022, Appendix A))*

$$dq_t = 2q_t \left( \frac{1 - \frac{\sqrt{2}}{3\pi} q_t^{1/2}}{t} - 1 \right) dt + 2\sqrt{2} q_t \, dB_t. \tag{4.7}$$

The above statement holds only when $t_0 > 0$, and there is a interesting technicality that must be resolved to interpret what happens as $t \to 0^+$. In particular, the Markov chain is not time homogeneous, and the limiting SDE has a singularity at $t = 0$. The contribution of the singularity needs to be controlled in order to establish convergence for all $t \geq 0$. Furthermore, due to the singularity issue, it is also unclear what the initial condition of $q_t$ should be.

In our simulations for Figure 1, we addressed the time singularity by shifting the time evaluation of $\frac{1}{t}$ to the next step of $\frac{1}{t + \Delta_t}$, where $\Delta_t > 0$ is the time step size. More precisely, we first consider the log version of $r_t = \log q_t$

$$dr_t = -2 \left( 1 - \frac{1 - \frac{\sqrt{2}}{3\pi} \exp(\frac{r_t}{2})}{t} \right) dt + 2\sqrt{2} dB_t. \tag{4.8}$$

Then we choose the following discretization

$$r_{t+\Delta_t} = r_t - 2 \left( 1 - \frac{1 - \frac{\sqrt{2}}{3\pi} \exp(\frac{r_t}{2})}{t + \Delta_t} \right) \Delta_t + 2\sqrt{2} \xi_t \sqrt{\Delta_t}, \tag{4.9}$$

where $\xi_t \sim N(0, 1)$. For initial conditions, we also noticed that since the initial correlation must be contained in the interval $[-1, 1]$, the end result was not very sensitive to the choice of $r_0$.

*Proof of Theorem 4.1.* Firstly, we will introduce the definitions

$$K_{p,r}(\rho) = \mathbb{E}\, \varphi_s(g)^p \varphi_s(\hat{g})^r \,, \tag{4.10}$$

where $g, w \sim N(0,1)$ and we define $\hat{g} = \rho g + qw$ with $q = \sqrt{1-\rho^2}$. We will also use the short hand notation to write $K_p := K_{p,p}$. Here we will recall several formulae calculated in Cho & Saul (2009) and (Li et al., 2022, Lemma B.4)

$$
\begin{aligned}
K_0(\rho) &= \mathbb{E}\, \mathbb{1}_{\{g>0\}}\mathbb{1}_{\{\rho g + qw > 0\}} = \frac{\arccos(-\rho)}{2\pi}\,, \\
K_1(\rho) &= \mathbb{E}\, \varphi(g)\varphi(\rho g + qw) = \frac{q + \rho\arccos(-\rho)}{2\pi}\,, \\
K_2(\rho) &= \mathbb{E}\, \varphi(g)^2\varphi(\rho g + qw)^2 = \frac{3\rho q + \arccos(-\rho)(1 + 2\rho^2)}{2\pi}\,, \\
K_{3,1}(\rho) &= \mathbb{E}\, \varphi(g)^3\varphi(\rho g + qw) = \frac{q(2 + \rho^2) + 3\arccos(-\rho)\rho}{2\pi}\,..
\end{aligned}
\tag{4.11}
$$

Furthermore, we will define

$$M_2 := \mathbb{E}\,[c\varphi(g)^2 - 1]^2 = 5\,. \tag{4.12}$$

Using the steps of (Li et al., 2022, Proposition B.8), we can establish an approximate Markov chain

$$\rho_{\ell+1} = cK_1(\rho_\ell) + \frac{\widehat{\mu}_r(\rho_\ell)}{n} + \frac{\sigma_r(\rho_\ell)\xi_\ell}{\sqrt{n}} + O(n^{-3/2})\,, \tag{4.13}$$

where $\xi_\ell$ are iid with zero mean and unit variance, and

$$
\begin{aligned}
\mu_r(\rho_\ell) &= \mathbb{E}[\widehat{\mu}_r(\rho_\ell)|\rho_\ell] = \frac{c}{4}\left[K_1(c^2K_2 + 3M_2 + 3) - 4cK_{3,1}\right]\,, \\
\sigma_r^2(\rho_\ell) &= \frac{c^2}{2}\left[K_1^2(c^2K_2 + M_2 + 1) - 4cK_1K_{3,1} + 2K_2\right]\,,
\end{aligned}
\tag{4.14}
$$

and we write $K_. = K_.(\rho_\ell)$.

Here we use the big $O(f(n,\ell))$ to denote a random variable $X$ such that for all $p \geq 1$

$$\frac{\mathbb{E}|X|^p}{f(n,\ell)^p} \leq C_p < \infty\,, \tag{4.15}$$

for some constants $C_p > 0$ independent of $n$ and $\ell$.

In view of the SDE convergence theorem Proposition A.7, if we eventually reach an SDE, we will only need to keep track of the expected drift $\mu_r$ instead of the random drift. We can then Taylor expand the coefficients in terms of $\rho_\ell$ about $\rho_\ell = 1$ (from the negative direction) using SymPy (Meurer et al., 2017), which translates to the following update rule

$$
\begin{aligned}
\rho_{\ell+1} = \rho_\ell &+ \frac{2\sqrt{2}}{3\pi}(1 - \rho_\ell)^{3/2} + \frac{\sqrt{2}}{30\pi}(1 - \rho_\ell)^{5/2} \\
&+ \frac{1}{n}\left(-2(1 - \rho_\ell) + \frac{4\sqrt{2}}{\pi}(1 - \rho_\ell)^{3/2} + 3(1 - \rho_\ell)^2 - \frac{73\sqrt{2}}{15\pi}(1 - \rho_\ell)^{5/2}\right) \\
&+ \frac{\xi_\ell}{\sqrt{n}}\left(2\sqrt{2}(1 - \rho_\ell) - \frac{56}{15\pi}(1 - \rho_\ell)^{3/2}\right) + O((1 - \rho_\ell)^4 + n^{-3/2})\,.
\end{aligned}
\tag{4.16}
$$

We note up to this point, a similar approach was taken in Jakub & Nica (2023). However, we will diverge here by consider the following scaling

$$q_\ell = \ell^2(1 - \rho_\ell)\,. \tag{4.17}$$

The choice of $\ell^2$ scale is motivated by the infinite-width limit, where $(1 - \rho_\ell)$ shown to be of order $O(\ell^{-2})$ in (Li et al., 2022, Appendix E). This implies the following update

$$
\begin{aligned}
\frac{\ell^2}{(\ell+1)^2} q_{\ell+1} = q_\ell &- \ell^2 \frac{2\sqrt{2}}{3\pi}(1-\rho_\ell)^{3/2} - \ell^2 \frac{\sqrt{2}}{30\pi}(1-\rho_\ell)^{5/2} \\
&- \frac{\ell^2}{n}\left(-2(1-\rho_\ell) + \frac{4\sqrt{2}}{\pi}(1-\rho_\ell)^{3/2} + 3(1-\rho_\ell)^2 - \frac{73\sqrt{2}}{15\pi}(1-\rho_\ell)^{5/2}\right) \\
&+ \frac{\xi_\ell \ell^2}{\sqrt{n}}\left(2\sqrt{2}(1-\rho_\ell) - \frac{56}{15\pi}(1-\rho_\ell)^{3/2}\right) + O(\ell^{-8}q_\ell^4 + \ell^{-2}n^{-3/2}).
\end{aligned}
\tag{4.18}
$$

Next, we will drop all higher order terms in $\ell$, then using the fact that $\frac{\ell^2}{(\ell+1)^2} = 1 - \frac{2}{\ell} + O(\ell^{-2})$, we can write

$$
q_{\ell+1} = q_\ell(1 + 2\ell^{-1}) - \frac{2\sqrt{2}}{3\pi}\frac{q_\ell^{3/2}}{\ell} - \frac{2q_\ell}{n} + \frac{\xi_\ell}{\sqrt{n}}2\sqrt{2}q_\ell + O(\ell^{-2} + n^{-1/2}\ell^{-1}),
\tag{4.19}
$$

where we dropped $\ell^{-2}n^{-3/2}$ in the big $O$ since it gets dominated by $\ell^{-2}$.

Choosing the time scaling $t = \frac{\ell}{n}$ then gives us

$$
q_{\ell+1} = q_\ell + \frac{2}{n}q_\ell\left(\frac{1 - \frac{\sqrt{2}}{3\pi}q_\ell^{1/2}}{t} - 1\right) + 2\sqrt{\frac{2}{n}}q_\ell\,\xi_\ell + O(t^{-2}n^{-2} + t^{-1}n^{-3/2}).
\tag{4.20}
$$

Finally, we will use the Markov chain convergence result to an SDE result Proposition A.7, which leads to the desired SDE for $t \geq t_0 > 0$

$$
dq_t = 2q_t\left(\frac{1 - \frac{\sqrt{2}}{3\pi}q_t^{1/2}}{t} - 1\right)dt + 2\sqrt{2}q_t\,dB_t.
\tag{4.21}
$$

$\square$

## 5 Discussion

In this section, we provide some discussion on the potential impact of this work, from both a practical and a theoretical point of view.

**Stable Training via Depth Scaling.** Martens et al. (2021) make the key observation that as depth increases, the increased instability of training dynamics is due to the key role of nonlinear activation functions. Since then, it is better understood that to achieve stable training in large depth, it is necessary to weaken the nonlinearities of each layer (Noci et al., 2023; Bordelon et al., 2023; Yang et al., 2023). Our results observe a key connection between two strategies, either weakening the activation function, or weakening the entire layer via skip connections directly. From a practical point of view, since both the shaping and ResNet approaches can lead to the same covariance ODE at initialization, we understand that the key to preventing unstable gradients is weakening nonlinearities. To choose between the two regimes, it is therefore left to study the role of weakening the weight matrix or otherwise, and how this affects training dynamics. In particular, we note that the shaped limit admits feature learning without modifying the scaling (Hanin & Nica, 2019a), which is fundamentally different than the $\mu$P regime (Yang & Hu, 2021).

**Analysis of Normalization Methods.** Since the correlation Markov chain has large jumps, and quickly to a degenerate fixed point, it is intuitive to assume this chain does not admit a continuous time limit, and therefore difficult to analyze from an analytical approach. Indeed, this is the approach taken by Meterez et al. (2023), which yielded one of the first analysis of normalization methods in a deep network. However, our result provides a counter-intuitive understanding: it remains possible to analyze seemingly large discrete jumps in a Markov chain if rescaled appropriately. In our case, since the Markov chain converges quickly to

the fixed point at $\rho = 1$, zooming in around the fixed point as a function of time (or layer $\ell$) allows us to view the dynamics at the correct scale. This overcomes a previously known technical hurdle, and opens up the possibility of analyzing normalization methods, which is still lacking theoretical progress. More specifically, we would like to understand how normalization methods may help or hurt performance in practice, and how to best choose and tune these methods given their many variants, all of which are now made easier by our scaling approach.

**Foundation for Studying Training Dynamics.** Until recently, almost all of the work on infinite-depth neural networks remain at initialization. This is not due to a lack of attempts, but rather due to a lack of mathematical techniques available. In particular, we also emphasize the seminal work of neural tangent kernels for training dynamics (Jacot et al., 2018) is entirely built on the same techniques studying initialization (Neal, 1995; Lee et al., 2018). For this reason, it is important to slowly yet firmly build up a foundation of theoretical results, that understands as much structure as possible at initialization. To this goal, the key distinction from the infinite-width regime is that each layer must be viewed as an infinitesimal discretization of a continuous "layer time." This approach is what helped yield some of the first characterization of training dynamics for infinite-depth ResNets (Bordelon et al., 2023; Yang et al., 2023). In fact, any theory of training dynamics must also account for this infinitesimal treatment of layers, otherwise the limit cannot possibly be stable. Therefore, we view this line of work as building towards a theory of training dynamics, and eventually generalization as well.

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

# A  Background on Markov Chain Convergence to SDEs

In this section, we will review the main technical results used for Markov chain convergence to SDEs. In particular, we will consider piecewise constant interpolations of Markov chains in continuous time defined by $V^{(n)}_{\lfloor tn \rfloor}$, which converges to a continuous process $V_t$. Due to measurability concerns of the Markov chain with jumps, in order to define weak convergence on the space of processes, we need to define a precise space along with its equipped topology first introduced by Skorohod. We will mostly follow the references Kallenberg (2021); Ethier & Kurtz (2009); Stroock & Varadhan (1997) in the rest of this section, and the content will be essentially a rephrasing of Appendix A in Li et al. (2022).

To start, we will let $S$ be a complete separable metric space, where the stochastic processes takes value in (in our case, it's the upper triangular entries $\mathbb{R}^{m(m+1)/2}$). Let $D_{\mathbb{R}_+, S}$ be the space of càdlàg functions, i.e. functions that are right continuous with left limits, from $\mathbb{R}_+$ to $S$. For $x_n \in \mathbb{R}_+$, we use $x_n \xrightarrow{ul} x$ to denote locally uniform convergence, or uniform convergence on compact subsets of $\mathbb{R}_+$. We also consider a class of bijections $\lambda$ on $\mathbb{R}_+$ such that $\lambda$ is strictly increasing, and $\lambda_0 = 0$. We define **Skorohod convergence**, written as $x_n \xrightarrow{s} x$ on $D_{\mathbb{R}_+, S}$, if there exists a sequence of such bijections $\lambda_n$ such that

$$\lambda_n \xrightarrow{ul} \mathrm{Id}, \quad x_n \circ \lambda_n \xrightarrow{ul} x. \tag{A.1}$$

Intuitively, this allows us to side step any discontinuities in the notion of convergence, as $\lambda_n$ is allowed to perform a "time change" so we always land on the "correct side" of the discontinuity.

Importantly, we note that $D_{\mathbb{R}_+, S}$ with the above notion of convergence forms a well behaved probability space.

**Theorem A.1** (Theorem A5.3, Kallenberg (2021)). *For any separable complete metric space $S$, there exists a topology $\mathcal{T}$ on $D_{\mathbb{R}_+, S}$ such that*

 *(i) $\mathcal{T}$ induces the Skorohod convergence $x_n \xrightarrow{s} x$,*

 *(ii) $D_{\mathbb{R}_+, S}$ is Polish (separable completely metrizable topological space) under $\mathcal{T}$,*

 *(iii) $\mathcal{T}$ generates the Borel $\sigma$-field generated by the evaluation maps $\pi_t$, $t \geq 0$, where $\pi_t(x) = x_t$.*

To be fully self-contained, we will also introduce the definitions of a Feller semi-group, generator, and the core. Let $S$ be a locally compact separable metric space, and let $C_0 = C_0(S)$ be the space of continuous functions that vanishes at infinity, equipped with the sup norm, hence making $C_0$ a Banach space. We say $T : C_0 \to C_0$ is a **positive contraction operator** if for all $0 \leq f \leq 1$, we have that $0 \leq Tf \leq 1$. A semi-group $(T_t)$ of positive contraction operators on $C_0$ is called a **Feller semi-group** if it also satisfies

$$\begin{aligned} T_t C_0 &\subset C_0, \quad t \geq 0, \\ T_t f(x) &\to x, \text{ as } t \to 0, \quad f \in C_0, x \in S. \end{aligned} \tag{A.2}$$

Let $\mathcal{D} \subset C_0$ and $A : \mathcal{D} \to C_0$. We say the pair $(A, \mathcal{D})$ is a **generator** of the semi-group $(T_t)$ if $\mathcal{D}$ is the maximal set such that

$$\lim_{t \to 0} \frac{T_t f - f}{t} = Af. \tag{A.3}$$

We say an operator $A$ with domain $\mathcal{D}$ on a Banach space $B$ is **closed**, if its graph $G = \{(f, Af) : f \in \mathcal{D}\}$ is a closed subset of $B \times B$. If the closure of $G$ is the graph of an operator $\overline{A}$, we say that $\overline{A}$ is the **closure** of $A$.

We say a linear subspace $D \subset \mathcal{D}$ is a **core** of $A$, if the closure of $A|_D$ is $A$. Intuitively, all important properties of $A$ can be recovered via a limit point of $Af_n$, for $f_n \in D$. Furthermore, if $(A, \mathcal{D})$ is a geneator of a Feller semi-group, every dense invariant subspace $D \subset \mathcal{D}$ is a core of $A$ (Kallenberg, 2021, Proposition 17.9). In particular, the core we will work with is the space $C_0^\infty$ of smooth functions that vanishes at infinity.

The following is a sufficient condition for a semi-group to be Feller.

**Theorem A.2** (Section 8, Theorem 2.5, Ethier & Kurtz (2009)). *Let $a^{ij} \in C^2(\mathbb{R}^d)$ with $\partial_k \partial_\ell a^{ij}$ be bounded for all $i, j, k, \ell \in [d]$, and let $b : \mathbb{R}^d \to \mathbb{R}^d$ be Lipschitz. Then the generator defined by the elliptic operator*

$$Af = \frac{1}{2} \sum_{i,j=1}^{d} a^{ij} \partial_i \partial_j f + \sum_{i=1}^{d} b^i \partial_i f, \tag{A.4}$$

*generates a Feller semi-group on $C_0$.*

Next, we will state a set of equivalent conditions for convergence of Feller processes.

**Theorem A.3** (Theorem 17.25, Kallenberg (2021)). *Let $X, X^1, X^2, X^3, \cdots$ be Feller processes in $S$ with semi-groups $(T_t), (T_{n,t})$ and generators $(A, \mathcal{D}), (A_n, \mathcal{D}_n)$, respectively, and fix a core $D$ for $A$. Then these conditions are equivalent:*

*(i) for any $f \in D$, there exists some $f_n \in \mathcal{D}_n$ with $f_n \to f$ and $A_n f_n \to Af$,*

*(ii) $T_{n,t} \to T_t$ strongly for each $t > 0$,*

*(iii) $T_{n,t} f \to T_t f$ for every $f \in C_0$, uniformly for bounded $t > 0$,*

*(iv) $X_0^n \xrightarrow{d} X_0$ in $S \Rightarrow X^n \xrightarrow{d} X$ in the Skohorod topology of $D_{\mathbb{R}_+, S}$.*

We note that it is common to choose the core $D = C_0^\infty$, and that checking the first condition is sufficient for convergence in the Skorohod topology. The next result will help us apply the above criterion to continuous time interpolated Markov chains.

**Theorem A.4** (Theorem 17.28, Kallenberg (2021)). *Let $Y^1, Y^2, Y^3, \cdots$ be discrete time Markov chains in $S$ with transition operators $U_1, U_2, U_3, \cdots$, and let $X$ be a Feller process with semi-group $(T_t)$ and generator $A$. Fix a core $D$ for $A$, and let $0 < h_n \to 0$. Then conditions $(i) - (iv)$ of Theorem A.3 remain equivalent for the operators and processes*

$$A_n = h_n^{-1}(U_n - I), \quad T_{n,t} = U_n^{\lfloor t/h_n \rfloor}, \quad X_t^n = Y_{\lfloor t/h_n \rfloor}^n. \tag{A.5}$$

Here we note that in our applications, we will always choose $h_n = n^{-2p}$, which is essentially dependent on the width of the neural network.

At this point, we still need to check that the generators $A_n$ converges to $A$ with respect to the core $D = C_0^\infty$. For this goal, we will use a lemma from Stroock & Varadhan (1997). Here we will define $\Pi_n(x, dy)$ to be the Markov transition kernel of $Y^n$, and let

$$a_n^{ij}(x) = \frac{1}{h_n} \int_{|y-x| \leq 1} (y_i - x_i)(y_j - x_j) \, \Pi_n(x, dy),$$

$$b_n^i(x) = \frac{1}{h_n} \int_{|y-x| \leq 1} (y_i - x_i) \, \Pi_n(x, dy), \tag{A.6}$$

$$\Delta_n^\epsilon(x) = \frac{1}{h_n} \Pi_n(x, \mathbb{R}^d \setminus B(x, \epsilon)).$$

**Lemma A.5** (Lemma 11.2.1, Stroock & Varadhan (1997)). *The following two conditions are equivalent:*

*(i) For any $R > 0, \epsilon > 0$ we have that*

$$\lim_{n \to \infty} \sup_{|x| \leq R} \|a_n(x) - a(x)\|_{op} + |b_n(x) - b(x)| + \Delta_n^\epsilon(x) = 0, \tag{A.7}$$

*(ii) For each $f \in C_0^\infty(\mathbb{R}^d)$, we have that*

$$\frac{1}{h_n} A_n f \to Af, \tag{A.8}$$

*uniformly on compact sets of $\mathbb{R}^d$, where $A$ is defined as (A.3).*

We will also need to weaken the definition slightly for non-Lipschitz coefficients.

**Definition A.6.** *We say a sequence of processes $X^n$ **converge locally** to $X$ in the Skorohod topology if for any $r > 0$, we define the following stopping times*

$$\tau^n := \{t \geq 0 : |X_t^n| \geq r\}, \quad \tau := \{t \geq 0 : |X_t| \geq r\}, \tag{A.9}$$

*and we have that $X_{t \wedge \tau^n}^n$ converge to $X_{t \wedge \tau}$ in the Skorohod topology.*

Finally, to summarize everything into a useful form for this paper, we will state the following proposition.

**Proposition A.7** (Convergence of Markov Chains to SDE, Proposition A.6, Li et al. (2022))**.** *Let $Y^n$ be a discrete time Markov chain on $\mathbb{R}^N$ defined by the following update for $p, \delta > 0$*

$$Y_{\ell+1}^n = Y_\ell^n + \frac{\widehat{b}_n(Y_\ell^n, \omega_\ell^n)}{n^{2p}} + \frac{\sigma_n(Y_\ell^n)}{n^p} \xi_\ell^n + O(n^{-2p-\delta}), \tag{A.10}$$

*where $\xi_\ell^n \in \mathbb{R}^N$ are iid random variables with zero mean, identity covariance, and moments uniformly bounded in $n$. Furthermore, $\omega_\ell^n$ are also iid random variables such that $\mathbb{E}[\widehat{b}_n(Y_\ell^n, \omega_\ell^n)|Y_\ell^n = y] = b_n(y)$ and $\widehat{b}_n(y, \omega_\ell^n)$ has uniformly bounded moments in $n$. Finally, $\sigma_n$ is a deterministic function, and the remainder terms in $O(n^{-2p-\delta})$ have uniformly bounded moments in $n$.*

*Suppose $b_n, \sigma_n$ are uniformly Lipschitz functions in $n$ and converges to $b, \sigma$ uniformly on compact sets, then in the limit as $n \to \infty$, the process $X_t^n = Y_{\lfloor tn^{2p} \rfloor}^n$ converges in distribution to the solution of the following SDE in the Skorohod topology of $D_{\mathbb{R}_+, \mathbb{R}^N}$*

$$dX_t = b(X_t)\, dt + \sigma(X_t)\, dB_t, \quad X_0 = \lim_{n \to \infty} Y_0^n. \tag{A.11}$$

*Suppose otherwise $b_n, \sigma_n$ are only locally Lipschitz (but still uniform in $n$), then $X^n$ converges locally to $X$ in the same topology (see Definition A.6). More precisely, for any fixed $r > 0$, we consider the stopping times*

$$\tau^n := \inf\{t \geq 0 : |X_t^n| \geq r\}, \quad \tau := \inf\{t \geq 0 : |X_t| \geq r\}, \tag{A.12}$$

*then the stopped process $X_{t \wedge \tau^n}^n$ converges in distribution to the stopped solution $X_{t \wedge \tau}$ of the above SDE in the same topology.*

## B  Technical Lemmas for Shaped Activations

Here we will recall and slightly modify a collection of definitions and technical results from Appendix B and C of Li et al. (2022), which are related to the shaped activation that we will use in the main theorems. To start, we will let $\varphi(x) = \max(x, 0)$, and recall the ReLU-like activation function as

$$\varphi_s(x) = s_+ \max(x, 0) + s_- \min(x, 0) = s_+ \varphi(x) - s_- \varphi(x). \tag{B.1}$$

Let $g \sim \mathcal{N}(0, 1)$, then we will restate Li et al. (2022, Lemma B.3, B.6)

$$\mathbb{E}\,\varphi(g) = \frac{1}{\sqrt{2\pi}}, \quad \mathbb{E}\,\varphi(g)^2 = \frac{1}{2}, \quad \mathbb{E}\,\varphi(g)^4 = \frac{3}{2}.$$

$$\mathbb{E}\,\varphi_s(g) = \frac{s_+ - s_-}{\sqrt{2\pi}}, \quad \mathbb{E}\,\varphi_s(g)^2 = \frac{s_+^2 + s_-^2}{2}, \quad \mathbb{E}\,\varphi_s(g)^4 = \frac{3}{2}(s_+^4 + s_-^4) \quad . \tag{B.2}$$

We will also recall the definitions

$$\bar{J}_{p,r}(\rho) := \mathbb{E}\,\varphi(g)^p \varphi(\hat{g})^r, \quad K_{p,r}(\rho) := \mathbb{E}\,\varphi_s(g)^p \varphi_s(\hat{g})^r, \tag{B.3}$$

where $g, w$ are iid $\mathcal{N}(0, 1)$ and we define $\hat{g} = \rho g + qw$ with $q = \sqrt{1 - \rho^2}$. We will also use the short hand notation to write $\bar{J}_p := \bar{J}_{p,p}, K_p := K_{p,p}$.

Here we recall from Cho & Saul (2009) and Li et al. (2022, Lemma B.7) that

$$\bar{J}_1(\rho) = \frac{\sqrt{1-\rho^2} + (\pi - \arccos \rho)\rho}{2\pi}, \quad K_1(\rho) = (s_+^2 + s_-^2)\bar{J}_1(\rho) - 2s_+s_-\bar{J}_1(-\rho). \tag{B.4}$$

Next we will slightly modify a Taylor expansion result from (Li et al., 2022, Lemma C.1)

**Lemma B.1** (Taylor Expand Shaping Correlation). *Recall* $\varphi_s(x) = s_+ \max(x,0) + s_- \min(x,0)$. *Let* $s_\pm = 1 + \frac{c_\pm}{n^p}$ *for* $p > 0$, *then we have the following Taylor expansion*

$$cK_1(\rho) = \rho + \frac{\nu(\rho)}{n^{2p}} + O(n^{-3p}), \quad \nu(\rho) = \frac{(c_+ - c_-)^2}{2\pi} \left(\sqrt{1-\rho^2} + \rho \arccos \rho\right). \tag{B.5}$$

*Proof.* We start by expanding the formula for $K_1(\rho)$ to get

$$cK_1(\rho) = \frac{2}{s_+^2 + s_-^2} \frac{1}{2\pi} \left((s_+^2 + s_-^2)\left(\sqrt{1-\rho^2} + (\pi - \arccos \rho)\rho\right) - 2s_+s_-\left(\sqrt{1-\rho^2} - (\arccos \rho)\rho\right)\right). \tag{B.6}$$

At this point, we can plug in $s_\pm = 1 + \frac{c_\pm}{n^p}$, and Taylor expanding gives us

$$\begin{aligned}
cK_1(\rho) = {} & \frac{\rho \arccos(\rho)}{\pi} + \frac{\rho(\pi - \arccos(\rho))}{\pi} \\
& + (n^{-p})^2 \left(\frac{-\rho c_+^2 \arccos(\rho) + 2\rho c_+ c_- \arccos(\rho) - \rho c_-^2 \arccos(\rho)}{2\pi}\right. \\
& \left. + \frac{c_+^2 \sqrt{1-\rho^2} - 2c_+ c_- \sqrt{1-\rho^2} + c_-^2 \sqrt{1-\rho^2}}{2\pi}\right) \\
& + O\left(\left(n^{-p}\right)^3\right).
\end{aligned} \tag{B.7}$$

Simplifying the expressions gives us the desired result of

$$cK_1(\rho) = \rho + \frac{\nu(\rho)}{n^{2p}} + O(n^{-3p}). \tag{B.8}$$

$\square$

Similar to Li et al. (2022, Lemma C.2), we can also approximate the fourth moment of shaped activation functions via the fourth moment of Gaussians.

**Lemma B.2** (Fourth Moment Approximation). *Considers the jointly Gaussian random variables*

$$[g^\alpha]_{\alpha=1}^4 \sim \mathcal{N}\left(0, [\rho^{\alpha\beta}]_{\alpha,\beta=1}^4\right), \tag{B.9}$$

*where* $\rho^{\alpha\alpha} = 1$ *for all* $\alpha$. *Let* $\varphi_s(x) = s_+ \max(x,0) + s_- \min(x,0)$ *with coefficients* $s_\pm = 1 + \frac{c_\pm}{n^p}$, *then we have*

$$\mathbb{E} \prod_{\alpha=1}^4 \varphi_s(g^\alpha) = \mathbb{E} \prod_{\alpha=1}^4 g^\alpha + O(n^{-p}) = \rho^{12}\rho^{34} + \rho^{13}\rho^{24} + \rho^{14}\rho^{23} + O(n^{-p}). \tag{B.10}$$

*Proof.* Observe that we can write $\varphi_s(x)$ as a perturbation of identity

$$\varphi_s(x) = x + \frac{1}{n^p}(c_+\varphi(x) - c_-\varphi(-x)) = x + O(n^{-p}). \tag{B.11}$$

This means we can approximate the product of shaped activations without the activation

$$\prod_{\alpha=1}^4 \varphi_s(g^\alpha) = \prod_{\alpha=1}^4 g^\alpha + O(n^{-p}). \tag{B.12}$$

Finally, we can use the Isserlis Theorem to write

$$\mathbb{E} \prod_{\alpha=1}^{4} g^{\alpha} = \mathbb{E} g^1 g^2 \mathbb{E} g^3 g^4 + \mathbb{E} g^1 g^3 \mathbb{E} g^2 g^4 + \mathbb{E} g^1 g^4 \mathbb{E} g^2 g^3 = \rho^{12}\rho^{34} + \rho^{13}\rho^{24} + \rho^{14}\rho^{23}, \tag{B.13}$$

which is the desired result.

$$\square$$

Furthermore, we can characterize the covariance of the following random variables

$$R^{\alpha\beta} = \frac{1}{\sqrt{n}} \sum_{i=1}^{n} \left[ c\varphi_s(g_i^{\alpha})\varphi_s(g_i^{\beta}) - cK_1(\rho^{\alpha\beta}) \right], \tag{B.14}$$

where the random vector $[g_i^{\alpha}, g_i^{\beta}]$ are iid copies of the same Gaussian vector. This follows from a modification of Li et al. (2022, Lemma C.3).

**Lemma B.3** (Covariance of $R^{\alpha\beta}$). *Let $R^{\alpha\beta}$ be defined as above. Then if $s_{\pm} = 1 + \frac{c_{\pm}}{n^p}$ for the shaped activation, we have that*

$$\mathbb{E} R^{\alpha\beta} R^{\gamma\delta} = \rho^{\alpha\gamma}\rho^{\beta\delta} + \rho^{\alpha\delta}\rho^{\beta\gamma} + O(n^{-p}). \tag{B.15}$$

*Proof.* Firstly, we note since each entry of the sum in $R^{\alpha\beta}$ are iid, we will only need to compute the moments for a single entry. This means

$$\mathbb{E} R^{\alpha\beta} R^{\gamma\delta} = \mathbb{E} c^2 \left( \varphi_s(g_i^{\alpha})\varphi_s(g_i^{\beta}) - K_1(\rho^{\alpha\beta}) \right) \left( \varphi_s(g_i^{\gamma})\varphi_s(g_i^{\delta}) - K_1(\rho^{\gamma\delta}) \right). \tag{B.16}$$

At this point, we observe that $c = 1 + O(n^{-p})$, $K_1(\rho) = \rho + O(n^{-2p})$, and we can write

$$\mathbb{E} R^{\alpha\beta} R^{\gamma\delta} = \mathbb{E} \left( \varphi_s(g_i^{\alpha})\varphi_s(g_i^{\beta}) - \rho^{\alpha\beta} \right) \left( \varphi_s(g_i^{\gamma})\varphi_s(g_i^{\delta}) - \rho^{\gamma\delta} \right) + O(n^{-p}). \tag{B.17}$$

Finally, the fourth moment approximation Lemma B.2 gives us the desired result

$$\begin{aligned}
\mathbb{E} R^{\alpha\beta} R^{\gamma\delta} &= \rho^{\alpha\beta}\rho^{\gamma\delta} + \rho^{\alpha\gamma}\rho^{\beta\delta} + \rho^{\alpha\delta}\rho^{\beta\gamma} - \rho^{\alpha\beta}\rho^{\gamma\delta} - \rho^{\alpha\beta}\rho^{\gamma\delta} + \rho^{\alpha\beta}\rho^{\gamma\delta} + O(n^{-p}) \\
&= \rho^{\alpha\gamma}\rho^{\beta\delta} + \rho^{\alpha\delta}\rho^{\beta\gamma} + O(n^{-p}).
\end{aligned} \tag{B.18}$$

$$\square$$

