# OpenReview forum: "Differential Equation Scaling Limits of  Shaped and Unshaped Neural Networks"
_TMLR — Accepted by TMLR_

### Review · Reviewer_mCoA · 2023-11-26

**Summary Of Contributions:**

This paper contributes two ideas to the asymptotic analysis of infinitely wide and deep neural networks. In this line of work, it is common to analyze what happens to the "covariance" in each layer, that is the expectation of how the activations in layer L for input x1 covary with those for input x2. This type of analysis has been useful in understanding neural network hyperparameters including weight initializations, learning rates, activation functions, batch norm, and layer norm.

In the present paper, the authors first draw a connection between two previously-published analyses, showing that the ODE describing the depth-evolution of the covariance in a infinitely wide and deep ResNet (Hayou and Yang 2023) is the drift term in the SDE describing the depth-evolution of the covariance in a fully-connected multilayer perceptron with shaped ReLU activation (Li et al 2022). Secondly, the paper provides an analysis of the covariance for an unshaped ReLU MLP.

**Audience:**

Yes

**Claims And Evidence:**

Yes

**Requested Changes:**

I have no major requested changes.
Some typos:
- Section 1, in the description of the first contribution, "Furthermore in the limit requires the scale ratio..."
- Section 1.1 ends with "...description remains an open."

**Strengths And Weaknesses:**

Strengths:
- The paper is well written.
- The organization is clear.
- I appreciated the authors' efforts to provide intuition with the mathematics.
- The contributions are clearly specified in relation to previous work.


Weaknesses:
- As a non-expert in this particular field, I find it difficult to judge how useful and interesting these contributions are to this line of work. It is clear to me that they *could* be interesting to some researchers in this field, but I am not familiar enough with the area to be confident.
- It would be helpful to have a conclusion that summarizes why these contributions will be interesting to other researchers studying scaling limits and machine learning more generally.

---

> ### Author Response · Authors · 2024-01-26
> **Response**
>
> Thank you for the constructive feedback. We have taken your comments seriously and have added significant revisions to our manuscript. In particular, we want to highlight that an entirely new discussion section has been added to address the concerns on practical impacts of studying scaling limits.
>
> To summary our significant revision of the manuscript, we list the major changes below (as we have written for other reviewers as well):
> 1. We have added a table of notations in the main text.
>
> 2. In the background section, we have added an extensive discussion of the context for the theoretical results of both the covariance SDE and ODE, along with a comprehensive set of definitions for all the notations introduced.
>
> 3. After stating the result of Theorem 2.1, we have also added interpretations for the coefficients of the SDE.
>
> 4. We have significantly expanded the proof of Lemma 3.1, with references to some additional technical results in Appendix B, now making this manuscript entirely self-contained.
>
> 5. Proposition 3.4 now references known technical results in Appendix A, which is another self-contained section on the technical details of weak convergence with respect to the Skorohod topology.
>
> 6. We have added a full discussion section, providing additional commentary on the impact of this theoretical analysis, especially referencing the potential practical implications.
>
> We hope the current version of the article leads to a far more accessible read for a broader audience.
>
> Please let us know if you have any additional feedback, as we would be happy to continue improving our manuscript further with your help.

---

### Review · Reviewer_BR7V · 2023-11-26

**Summary Of Contributions:**

This paper relates between scaling limits of shaped and unshaped neural networks. Specifically, the authors show that certain unshaped ResNets converge to shaped MLP. In addition, the authors derive correlation results for unshaped MLPs.

**Audience:**

Yes

**Claims And Evidence:**

No

**Requested Changes:**

I believe the authors should strive to make their submission more accessible to general ML audience.

**Strengths And Weaknesses:**

Unfortunately, I am not an expert on this topic, so please regard my review accordingly.

Based on the authors' claims, the main strength of this submission is the relation between shaped and unshaped networks.

The main weakness is the exposition and clarity of writing. In particular, the paper is written for people working on this problem, and it is not accessible to a larger audience. Heavy use of jargon and prior results made it difficult to follow the general plot. In addition, the manuscript contains (or at least, what seems to be) several unfinished sentences. There is no conclusion section. The authors do not motivate their research problem, nor its significance and the impact of their results.

---

> ### Author Response · Authors · 2024-01-26
> **Response**
>
> Thank you for the constructive feedback. We have taken your review seriously, and provided a **significant revision** to our manuscript towards the goals of making this article more accessible to a broader ML audience. Please find a large amount of revised text highlighted in blue, as well as added sections.
>
> More specifically, these revisions are listed as follows (as we have written to other reviewers as well):
>
> 1. We have added a table of notations in the main text.
>
> 2. In the background section, we have added an extensive discussion of the context for the theoretical results of both the covariance SDE and ODE, along with a comprehensive set of definitions for all the notations introduced.
>
> 3. After stating the result of Theorem 2.1, we have also added interpretations for the coefficients of the SDE.
>
> 4. We have significantly expanded the proof of Lemma 3.1, with references to some additional technical results in Appendix B, now making this manuscript entirely self-contained.
>
> 5. Proposition 3.4 now references known technical results in Appendix A, which is another self-contained section on the technical details of weak convergence with respect to the Skorohod topology.
>
> 6. We have added a full discussion section, providing additional commentary on the impact of this theoretical analysis, especially referencing the potential practical implications.
>
> We believe our efforts have produced a far more pedagogical treatment of our theoretical and technical content.
>
> Please let us know if you have additional feedback for our revisions, as we would be happy to continue improving our article further with your help.

---

### Review · Reviewer_YWHM · 2023-12-24

**Summary Of Contributions:**

This article explores the behavior of neural networks with unshaped activations, where the activation function remains constant regardless of network size. Two main findings are presented:

1. Convergence of Two Architectures: It demonstrates that two different network architectures converge to the same limit when they are infinitely deep and wide. These architectures are:
   - A fully connected ResNet with a scaling factor inversely proportional to the square root of the network depth ($d^{-1/2}$).
   - A multilayer perceptron (MLP) with depth significantly less than width, employing a shaped ReLU activation scaled at a rate of $d^{-1/2}$.

2. Layerwise Correlation in Unshaped MLPs: For an unshaped MLP, the research derives a first-order asymptotic correction to the layerwise correlation. Specifically, if $\rho_{\ell}$ is the correlation at a certain layer $\ell$, then $q_t = \ell^2(1 - \rho_{\ell})$ with $t = \frac{\ell}{n}$ converges to a stochastic differential equation (SDE) with a singularity at $t = 0$.

These insights bridge the gap between shaped and unshaped neural network architectures. They also pave the way for investigating how normalization methods might be related to shaping activation functions.

**Audience:**

Yes

**Claims And Evidence:**

No

**Requested Changes:**

- Restate clearly Theorem 2.1 - with all definitions and notations, conclusions.
- Make in an appendix a list of definitions [when reading results, would be great to know where the symbols are defined]
- Restate clearly Theorem 2.2
- Would like to have a detailed derivation of Theorem 4.1 - The treatment of complicated mathematical concepts, especially the convergence of processes, suffers from the imprecision of definitions and constructions. A detailed proof from first principle is essential. Such clarity would not only improve the understanding of the subject, but also maintain the precision expected in mathematical presentation.
- The article would benefit from a clearer discussion of the significance and practical implications of its findings. In particular, it should clarify how these results influence our understanding of learning mechanisms and guide the selection of activation functions in neural networks. At present, the relevance and scope of these results for practise remain rather vague. A more explicit link between the theoretical results and their practical applications would greatly enhance the reader's understanding of the implications and relevance of the study in this area.

**Strengths And Weaknesses:**

Strengths:
Two key results -
1. Shaping Activation and ResNet connection: The authors note that the shaping of the activation function in neural networks is closely related to ResNet architectures. In particular, they note that the covariant ordinary differential equation (ODE) is merely the deterministic drift part of a broader covariant stochastic differential equation (SDE). Moreover, for the shaped multilayer perceptron (MLP), they find that if the scaled ratio $\frac{d}{n^{2p}}$ converges to a positive constant, where $p$ belongs to the interval $(0, \frac{1}{2})$, the covariance of the shaped MLP also converges to this same ODE. This result emphasises the connection between shaped activations and ResNet structures in neural networks.

2. Correlation Analysis in Unshaped MLP: The second result deals with the correlation dynamics within an unformed MLP. They derive the first-order asymptotic correction for the layer-wise correlation. This correction is introduced by rescaling the correlation $\rho_{\ell}$ in the layer $\ell$ using the term $q_{\ell} = \ell^2(1 - \rho_{\ell})$. Their analysis shows that this rescaled correlation is approximated by a stochastic differential equation (SDE), indicating a relationship between layer depth, width and correlation in unshaped MLPs.

These findings contribute to our understanding of the behavior of neural networks. They not only provide a link between shaped and unshaped network architectures, but also offer a new perspective on the dynamics of correlations in neural networks and the principles governing them.

Weakness:
- The complexity of the article arises from the extensive recourse to external references and the brevity of the proof, which presupposes knowledge of numerous previous results. This approach makes it difficult for readers to understand the arguments put forward. To improve comprehensibility, it would be beneficial to construct the theoretical objects with due precision, formulate the assumptions more clearly. Such a detailed presentation would help readers understand the complicated concepts and logical flow of the arguments, making the material more accessible and informative.

---

> ### Author Response · Authors · 2024-01-26
> **Response**
>
> Thank you for the detailed and constructive review, and we want to apologize for the delayed response. My schedules in the past month has not been very kind to me, and we wanted to provide an **extensive revision** as requested by you. We believe we have addressed all of your requests and concerns at this point, and we will detail these below.
>
> Here is a list of significant revisions in the updated manuscript, with many of the revisions highlighted in blue, along with three added sections:
>
> 1. We have added a table of notations in the main text.
>
> 2. In the background section, we have added an extensive discussion of the context for the theoretical results of both the covariance SDE and ODE, along with a comprehensive set of definitions for all the notations introduced.
>
> 3. After stating the result of Theorem 2.1, we have also added interpretations for the coefficients of the SDE.
>
> 4. We have significantly expanded the proof of Lemma 3.1, with references to some additional technical results in Appendix B, now making this manuscript entirely self-contained.
>
> 5. Proposition 3.4 now references known technical results in Appendix A, which is another self-contained section on the technical details of weak convergence with respect to the Skorohod topology.
>
> 6. We have added a full discussion section, providing additional commentary on the impact of this theoretical analysis, especially referencing the potential practical implications.
>
> We hope that this significant revision has now led to a far more pedagogical treatment of the theoretical and technical content, and will also now appeal to a far greater range of readers.
>
> Please let us know if you have any additional feedback, as we would be happy to revise our manuscript for further improvements.

---

### Decision · Action_Editor_WS2c · 2024-04-17

**Recommendation:** Accept as is

**Comment:**

This paper analyzes the infinite-width-and-depth limit of certain neural network architectures at initialization, finding connections between residual networks and shaped activations, as well as the leading-order correction to the asymptotic activation covariance in standard ReLU MLPs.

The reviewers generally appreciated the results and the intuition provided to help readers understand the findings. The derived SDEs provide a precise new basis for our understanding of the dynamics of signal propagation at initialization, and will surely be of interest to the community. As the authors note, conclusions about initialization do not necessarily carry over during training, but I agree that building a deep understanding at initialization is a prerequisite for the much more ambitious goal of characterizing training dynamics. All together, the scope and analysis seem entirely appropriate for TMLR.

The reviewers noted that the original submission was challenging to read for someone without expertise in the area. In the revision, the authors provided significantly more background, discussion, and details that make the paper considerably more approachable. As such, I believe the paper is now appropriate to publish, and recommend accepting the paper as is.

**Audience:**

Yes, some members of the community will be interested in this paper.

**Claims And Evidence:**

Yes, the claims are supported by convincing evidence.